# Hypothalamic Reactivity and Connectivity following Intravenous Glucose Administration

**DOI:** 10.3390/ijms24087370

**Published:** 2023-04-17

**Authors:** Joe J. Simon, Pia M. Lang, Lena Rommerskirchen, Martin Bendszus, Hans-Christoph Friederich

**Affiliations:** 1Department of General Internal Medicine and Psychosomatics, University Hospital Heidelberg, 69120 Heidelberg, Germany; 2Department of Neuroradiology, University Hospital Heidelberg, 69120 Heidelberg, Germany

**Keywords:** fMRI, glucose infusion, hypothalamus, intravenous glucose, energy homeostasis

## Abstract

Dysfunctional glucose sensing in homeostatic brain regions such as the hypothalamus is interlinked with the pathogenesis of obesity and type 2 diabetes mellitus. However, the physiology and pathophysiology of glucose sensing and neuronal homeostatic regulation remain insufficiently understood. To provide a better understanding of glucose signaling to the brain, we assessed the responsivity of the hypothalamus (i.e., the core region of homeostatic control) and its interaction with mesocorticolimbic brain regions in 31 normal-weight, healthy participants. We employed a single-blind, randomized, crossover design of the intravenous infusion of glucose and saline during fMRI. This approach allows to investigate glucose signaling independent of digestive processes. Hypothalamic reactivity and connectivity were assessed using a pseudo-pharmacological design and a glycemia-dependent functional connectivity analysis, respectively. In line with previous studies, we observed a hypothalamic response to glucose infusion which was negatively related to fasting insulin levels. The observed effect size was smaller than in previous studies employing oral or intragastric administration of glucose, demonstrating the important role of the digestive process in homeostatic signaling. Finally, we were able to observe hypothalamic connectivity with reward-related brain regions. Given the small amount of glucose employed, this points toward a high responsiveness of these regions to even a small energy stimulus in healthy individuals. Our study highlights the intricate relationship between homeostatic and reward-related systems and their pronounced sensitivity to subtle changes in glycemia.

## 1. Introduction

The body’s energy household is an intricate system balancing energy uptake with energy expenditure to achieve homeostasis. Food intake thus primarily depends on the interplay between the current energy status of the body, hedonic aspects of eating (i.e., taste, smell or social) and external factors such as food availability and time of day. The hypothalamus is located at the center of this system as it receives information about energy needs directly via circulating factors such as glucose, insulin, ghrelin, leptin, etc., as well as spinal and vagal afferents from the gut [1], but also about hedonic aspects through connections to dopaminergic mesocorticolimbic brain regions such as the ventral tegmental area, nucleus accumbens, and the posterior mediofrontal cortex [2]. These regions are involved in a wide array of goal-directed behaviors by predicting action-outcome contingencies and implementing control strategies to bias future choices toward options that have previously been perceived as pleasurable or rewarding (for example, the intake of one food type instead of another [3,4]). Although dopaminergic neurotransmission within this reward system is influenced by metabolic signals [5], work in animals has shown that food intake can take place even in the absence of dopaminergic mesolimbic neurotransmission [6]. Thus, the hypothalamus mediates food intake, which, in turn, is modulated by the dopaminergic reward system.

Alterations within or between these systems result in both over- and undernutrition and, in extreme cases, in eating disorders such as Anorexia Nervosa (AN) or obesity [7,8]. Specifically, whereas nutrient intake during hunger induces a consistent reduction of hypothalamic activity in normal-weight participants [9,10], both obesity and AN display blunt hypothalamic reactivity to sugar ingestion [11]. Furthermore, connectivity between the hypothalamus and reward, as well as gustatory-related brain regions, is impaired in both AN and obesity, pointing toward central processing of food stimuli that is disconnected from homeostatic signaling [11,12]. These results are in line with the common observation of dysfunctional neuronal processing in brain reward networks during exposure to food stimuli in both obesity and AN [13,14].

Taken together, homeostatic regulation and food intake are governed by a multitude of regulation processes. To better understand how certain processes promote food ingestion beyond or below nutritional needs, it may be worthwhile to study these systems in isolation. For instance, in a previous study, we controlled for the cephalic phase of digestion as well as cortical signals by infusing glucose directly into the stomach of healthy participants [11]. Here, we observed the expected hypothalamus deactivation as well as a concurrent glycemia-dependent connectivity with mesocorticolimbic reward networks. To extend these results, the present study aimed to isolate reward-related and metabolic systems during food intake even further by applying glucose via a mere intravenous infusion, thereby controlling for the entire digestive process by raising blood glucose levels directly.

Short-term hormonal changes after intravenous glucose infusion are expected to be similar to those observed after oral glucose ingestion [15]. Furthermore, a number of previous studies have employed the intravenous infusion of glucose to assess hypothalamic reactivity using fMRI in normal-weight and overweight participants [16,17,18,19]. Whereas one study failed to find hypothalamic reactivity following glucose infusion in nine healthy participants [18], others observed decreased activity in the hypothalamus [17,20]. These conflicting results can be, in part, attributed to differences in sample size, since oral intake and intragastric infusion of glucose induce a more pronounced deactivation of the hypothalamus then intravenous administration [17,21]. Furthermore, intravenous glucose infusion has been found to influence hypothalamus reactivity during emotional processing [16] and during the processing of visual food stimuli [19], suggesting that a state of hyperglycemia is sufficient to trigger hypothalamic-corticolimbic processing.

Our goal was to replicate previous observations in hypothalamus reactivity following an isolated rise in blood glucose levels, but to extend these results to inferences about glycemia-related brain-connectivity. As opposed to previous studies, where a continuous infusion of glucose throughout scanning was employed, we applied a singular and much smaller dose of glucose, individually calculated to achieve post-prandial levels of glycemia (approximately 150 mg/dL). To this end, healthy and normal-weight participants received both an intravenous glucose and saline solution during fMRI scanning. Hypothalamus reactivity was assessed using time-series analyses and glycaemia-dependent connectivity between the hypothalamus and mesocorticolimbic/gustatory brain areas and analyzed using a functional connectivity analysis. Finally, the relation between hypothalamic glucose reactivity and metabolic markers associated with glucose metabolism was assessed.

## 2. Results

### 2.1. Hormonal Satiety Parameters

Participants’ characteristics are given in Table 1. The results of the EDE-Q and BDI-II scales remained within normal range, while the results of the MWT-B point toward a mean IQ score of 107. Results of the blood sample analysis are given in Figure 1. The participants remained unaware of the type of liquid administered during each session (χ^2^ = 0.803, *p* = 0.370). We injected a mean dose of 9109.7 mg (±1254.4 mg) of glucose, corresponding to approximatively 45 mL of G20 solution (±6.27 mL). During the saline condition, we injected a mean of 45.1 mL of saline solution (±6.1 mL). All participants showed normal fasting glucose levels. Following the infusion of glucose, participants showed a significant increase in blood glucose levels (t_(30)_ = 8.2, *p* < 0.001) and insulin levels (t_(29)_ = 3.5, *p* < 0.001). Following the infusion of NaCl, we observed a significant decrease in both blood glucose and insulin levels (t_(29)_ = 4.6, *p* < 0.001 and t_(30)_ = 7.9, *p* < 0.001, respectively).

### 2.2. Changes in Hypothalamic Activation Induced by Glucose and Saline

As expected, a comparison of the hypothalamus Area Under the Curve (AUC) revealed a significant difference between glucose and saline infusion (t_(30)_ = 1.88, *p* = 0.035, see Figure 2a–c), showing a decrease in hypothalamus activity following glucose infusion. To assess the relation between hypothalamic reactivity and glucose metabolism, we correlated the hypothalamic AUC following glucose infusion with changes in blood glucose levels and changes in insulin levels. A negative AUC value indicates hypothalamic reactivity (i.e., corresponding to the expected deactivation of the hypothalamus following nutrient intake). During glucose infusion, no significant correlation between AUC values and changes in blood glucose levels (r = 0.339, *p* = 0.062) as well as changes in insulin levels (r = 0.285, *p* = 0.12) were found. However, insulin levels assessed prior to the infusion of glucose displayed a negative relation with the AUC of hypothalamus reactivity (pre-infusion insulin: r = −0.579, *p* < 0.001, see Figure 2d). Importantly, the individual BMI of participants did not correlate with pre-infusion insulin and blood glucose values, as well as with changes in blood glucose and insulin levels (Ps > 0.06). Scatterplots showing the (non-significant) relations between insulin/blood glucose levels and the AUC of hypothalamic reactivity are given in Figure 3. Since one of our participants had a BMI of 25.8 kg/m^2^, we repeated the comparison of the hypothalamus AUC with the exclusion of this participant; this did not have a significant impact on the results (t_(30)_ = 1.80, *p* = 0.04). Furthermore, the correlation between pre-infusion insulin levels and the AUC of hypothalamus reactivity remained significant (r = −0.584, *p* < 0.001).

### 2.3. Infusion-Type Dependent Functional Connectivity from the Hypothalamus to Reward-Related Brain Regions

We observed decreased functional connectivity from the hypothalamus to reward-related brain regions after glucose infusion when compared with saline infusion. Specifically, we observed decreased activation in the medial orbitofrontal cortex, anterior cingulate cortex, nucleus caudatus and putamen (see Table 2 and Figure 4a–c). The removal of one participant with a BMI of 25.8 kg/m^2^ did not significantly impact the observed whole-brain results.

## 3. Discussion

The overall aim of this study was to assess hypothalamic and mesocorticolimbic responsivity to changes in plasma glucose levels in healthy participants. To this end, we raised participants’ blood glucose levels via intravenous glucose injection with a dose just high enough to provoke a rise within a physiological range. We found this method to be both safe and feasible, and able to induce homeostatic processing in the brain. The results show a significant reduction of hypothalamus activity following glucose infusion when compared to saline infusion. Furthermore, we were able to show an infusion-type dependent functional connectivity between the hypothalamus and the brain reward network. Finally, we observed a significant correlation between hormonal parameters and activation in the hypothalamus.

Compared to previous studies, where glucose was administered either orally [9,10,22] or via a nasogastric tube [11,23], the hypothalamic reactivity we observed following intravenous glucose injection was smaller. Since the attenuation of hypothalamus activity is dependent on the amount of glucose ingested [9] and the concurrent rise in blood glucose concentration [20], this observation could be explained by the fact that the dose of intravenously administered glucose was lower than the amount of glucose used during gastric infusion [11,23]. Specifically, in a previous investigation [11], we administered 90 g of glucose and observed pre-post infusion differences in blood glucose levels of ±61.5 mg/dL, whereas in the current study, we administered ±9 g of glucose and observed a difference of ±18.4 mg/dL in blood glucose levels. However, it is also possible that the effect was smaller because by giving the glucose intravenously, we controlled for the digestive process (e.g., gastric distension). This is plausible considering the many afferents the hypothalamus receives from the digestive tract [24] and is in line with a previous observation of a smaller effect size of hypothalamic reactivity following intravenous glucose infusion when compared to oral glucose intake (for example, [17]). Importantly, when comparing hypothalamic reactivity in response to intragastric infusion with reactivity in response to intravenous infusion, the difference is linear in nature, i.e., a similar response pattern with a reduced effect size. Taken together, since we applied glucose just above the effective dose needed to produce a biological response, our results underline the sensitivity of hypothalamic glucose-sensing mechanisms. Further studies should investigate differences in neuronal activity following intragastric and intravenous infusion of the same amount of glucose, which could provide fundamental insights into pre-absorptive and post-absorptive glucose sensing.

We failed to observe a significant relation between changes in blood glucose and insulin levels following glucose infusion and activation in the hypothalamus. Since some nuclei of the hypothalamus (for example, the ventromedial nucleus) are central hubs for glucose-sensing mechanisms [25], and therefore essential in controlling glucose and energy metabolism [26], this result may come as a surprise. However, although the dose of glucose solution used for intravenous injection (mean dose of 9.1 g) was able to trigger a significant deactivation of the hypothalamus and increase in blood glucose levels, a dependency may only be statistically significant at higher blood glucose levels. Furthermore, plasma glucose levels and brain activity were measured at different time points. However, some previous studies assessing reactivity of the hypothalamus to glucose ingestion have also failed to observe a significant correlation [27,28].

Although we did not find any relation between blood glucose levels and the magnitude of hypothalamic reactivity, we were able to replicate previous findings of a relation between fasting plasma insulin concentrations and hypothalamic reactivity following glucose ingestion [29,30]. Insulin is an important hormone that signals satiety to the hypothalamus and is also involved in non-homeostatic aspects of food intake [31,32]. Insulin provides a satiety signal by decreasing the expression of orexigenic neuropeptides such as Neuropeptide Y [33]. Furthermore, animal studies have shown that insulin administered into the ventral tegmental area (VTA) inhibits palatable food eating [34] and decreases motivation to obtain sucrose [35]. This is in line with previous fMRI studies where a negative relation between plasma insulin levels and neuronal processing of food stimuli in the hypothalamus and mesocorticolimbic regions has been observed [36,37]. Taken together, our results add to the current conception of insulin as an important modulator of central control of feeding, glucose and energy homeostasis [38].

Finally, we found a glycemia-dependent connectivity between the hypothalamus and reward-related corticolimbic brain areas. This is in line with a previous study from our group, where we found a similar pattern of connectivity following intragastric infusion of a larger dose of glucose (90 g, [11]) and adds to the growing body of literature underpinning the close relation between hedonic and homeostatic brain networks during the control of food intake (for a review, see [39]). Furthermore, since we manipulated blood glucose levels directly, we were able to exclude the digestive process and the associated release of gastrointestinal satiety signals. A moderate hyperglycemic state appears to interact with dopaminergic neurons of the ventral tegmental area in a similar fashion to gastrointestinal hormones such as ghrelin, insulin, leptin and PYY [40].

### 3.1. Limitations

Our study has several limitations. Participants were a homogenous collective of young, healthy individuals, which limits inferences to the general population. In addition, about half of participants were hormonally active women. We did not control for the menstrual cycle in this study, which has been shown to have an influence on neuronal processing [41]. Accordingly, it would of great interest to analyze gender-related differences in hypothalamic reactivity in future studies. Furthermore, the MRI machine employed in this study was upgraded during our study (from a Siemens MAGNETOM Trio to a Siemens MAGNETOM Prisma). However, since the study was placebo-controlled and each subject underwent both scans in the same scanner, we believe the impact of this change to be limited. The obtained resolution of out MRI-sequence (2 mm^3^) may not be sufficient to study the reactivity of hypothalamic nuclei such as the arcuate or the ventromedial hypothalamic nuclei, which react differently to blood glucose levels [25]. In addition, since we only employed glucose in this study, the effects of macronutrients on hypothalamic processing could not be assessed. Because we did not assess additional peripheral hormone signals such as Ghrelin, Peptide Y or Glucagon-like peptide 1, our conclusions related to the isolated impact of glucose signaling on hypothalamic reactivity and connectivity have to be treated with caution. Since blood sampling during MRI scanning would have been disruptive, we did not measure the rise in blood glucose during scanning. Although we observed a significant rise in blood glucose levels assessed immediately after the scan, the mean rise in blood glucose levels was lower than the targeted 150 mg/dL post-prandial level. This was likely caused by the time lag between glucose infusion and post-scan blood sampling (approximatively 40 min).

### 3.2. Conclusions

Gut–brain signaling is an essential mechanism in the control of food intake. Homeostatic and reward-related brain regions work in concert to modulate appetite, satiation and satiety. We found that an isolated rise in blood glucose levels is sufficient to trigger typical, satiety-related responses in the hypothalamus. Furthermore, connectivity between the hypothalamus and reward-related mesocorticolimbic brain regions was similar to the pattern typically observed in response to satiety-inducing gastrointestinal hormones. Taken together, the intravenous infusion of glucose is a feasible experimental procedure to investigate gut–brain signaling independently from sensory aspects of food intake and digestive processing. We believe that the observed results may add to the current understanding of energy homeostasis and the crosstalk between homeostatic- and reward-related (mesocorticolimbic) brain networks.

## 4. Materials and Methods

### 4.1. Study Design and Test Group

We employed a single-blind, placebo-controlled, crossover experimental fMRI design of intravenous 20% glucose solution versus 0.9% sodium chloride solution as a placebo (see Figure 5 for a graphical depiction of the study design). In total, 34 subjects were recruited using flyer advertisements. To be included, participants had to be both physically and mentally healthy, able to undergo an MRI-scan and normal-weight (defined as a BMI of 19–25 kg/m^2^). However, one male participant was included despite having a BMI of 25.8 kg/m^2^, since his increased weight was caused by increased muscle mass. All participants were right-handed and were screened for medical and mental diseases by assessing their medical history and via the Structured Clinical Interview for DSM-IV [42]. Exclusion parameters were pregnancy, magnetic implants and claustrophobia. Three participants had to be excluded due to excessive head movement during scanning (exceeding 4 mm in each direction and ±3° of rotation), resulting in a final sample size of 31. Furthermore, seven participants had to be excluded from the connectivity analysis due to insufficient coverage of distal portions of the insular cortex or adjacent mesocorticolimbic regions (see methods for details regarding the acquisition procedure). Finally, due to technical reasons, insulin values could not be evaluated for one participant following glucose infusion, and blood glucose levels could not be assessed for one participant following saline infusion.

All participants filled out a number of self-report and demographic questionnaires and were asked about their eating and dieting behaviors. Specifically, we employed a Vocabulary-Based Test for the Assessment of Premorbid Intelligence (MWT-B; [43]), the Eating Disorder Examination Questionnaire (EDE-Q; [44]) and the Beck Depression Inventory (BDI-II; [45]). The EDE-Q is a well-established scale used to assess the range and severity of features associated with eating disorders. A commonly used cut-off score to assess the likely presence of an eating disorder is a score above 2.3 [46]; however, the practical value of using cut-off scores for the EDE-Q has been questioned [47]. The BDI-II is commonly used to assess symptoms of depression, where values above 14 point toward the presence of a clinical depression [48]. The MWT-B is designed as a short measure of verbal intelligence and has been found to be highly associated with general IQ assessments [49]. Based on data from previous validation studies, the sum score of the MWT-B can be used to approximate the IQ score of a participant (a sum score of 27 corresponds to an IQ score of 100) [43]. The medical ethics committee of the Medical Faculty Heidelberg at the Ruprecht-Karls-University in Heidelberg, Germany, approved this study (protocol number S-545/2019), and written informed consent was obtained from all participants.

### 4.2. Procedure

Participants underwent two MRI scans on separate days following a fasting period of at least 14 h. Scanning took place between 12:00 and 13:00, which was around lunchtime for most participants. They received placebo (NaCl: 0.9% sodium chloride in water, Ecoflac plus, B. Braun SE, Melsungen, Germany) during one appointment and verum (G20: 20% glucose in water, Ecoflac plus, B. Braun SE, Melsungen, Germany) during the other. The order was randomized, and subjects were blinded to the solution given at each time-point. Shortly before entering the scanner, we calculated the dosage of G20 necessary to raise blood glucose levels to approximately 150 mg/dL to simulate a physiological post-prandial level. In the first step, current blood glucose levels were determined using a glucometer (StatStrip Xpress Glucose Meter, Nova Biomedical Corporation, Waltham, MA, USA). In the second step, the required dose of G20 was calculated using the formula of the priming dose from the hyperglycemic clamp protocol, which uses current blood glucose levels (*x*) and body surface (*y*) as parameters [50]:$$dose\;[mg]=\left(150-x\right)\times \frac{9622}{125}\times y$$

Body surface was calculated using the formula according to Mosteller [51]. G20 dosage was calculated in milligrams but converted into milliliters for feasibility. For placebo, the corresponding amount of sodium chloride solution was injected. A peripheral venous catheter was placed in the median cubital vein to inject the test solution. To prevent a contamination of blood samples with residual G20 solution, blood samples used for the assessment of blood glucose and insulin levels before and after scanning were collected using 21-gauge Multifly needles (Sarstedt AG, Nümbrecht, Germany). Since a glucometer only provides a rough estimate of actual blood glucose values, blood samples were additionally analyzed in the laboratory to obtain a reliable estimation of blood glucose and insulin levels (see below). After a 5 min baseline fMRI scan, the solution was injected intravenously with a mean duration of 3 min. Following infusions, participants were scanned for another 24 min. After the scan, blood glucose and insulin levels were measured again. See Figure 5 for a graphical depiction of the study protocol.

### 4.3. fMRI Acquisition

Functional imaging was performed on a Tim Trio Scanner (Siemens Medical, Erlangen, Germany) using a 32-channel head coil. A gradient-recalled EPI sequence designed to reduce signal drop-out in the amygdala-hypothalamus region was employed [11,52,53,54]. Thirty-five axial slices, centered at the hypothalamus and aligned along the anterior commissure–posterior commissure (AC-PC), were acquired with a slice thickness of 2 mm without a gap. The field of view was 10.4 cm × 10.4 cm with a matrix size of 52 × 52 and a flip angle of 80°. Parallel imaging with a GRAPPA factor of two was used to enable both a TE of 30 ms and a TR of 2260 ms. The field of view of the echo-planar imaging (EPI) sequence provided reduced brain coverage with regions located dorsally from the corpus callosum left out.

In addition, a T1-weighted high-resolution anatomical image with 192 slices (1 × 1 × 1 mm voxel size, TR = 1900 ms, TE = 2.52 ms, flip angle = 9°, field of view = 25.6 cm × 25.6 cm) was acquired for anatomical reference.

### 4.4. fMRI Analysis

We analyzed fMRI data using the Blood Oxygen Level Dependent effect (BOLD). Specifically, the BOLD effect is a physiological response that occurs when there is an increase in neural activity in a particular area of the brain. This increase in neural activity leads to an increase in blood flow to that area, which, in turn, leads to an increase in the amount of oxygen that is delivered to the neurons in that region. The increased oxygenation of the blood changes its magnetic properties, which can be detected by the fMRI scanner. Accordingly, the BOLD signal is used as an indirect measure of brain activity, with increases in the signal indicating increased activity in the relevant brain regions [55].

We employed a ROI-based analysis to assess glucose-induced activation in brain regions related to homeostasis. For each participant and each condition (glucose, saline), the signal averages for 12 post-infusion time bins (each time bin or time window corresponds to 2 min of scanning) were compared with the baseline average (T0), a method comparable to differential regression analysis [56]. FMRI data were processed using SPM8-based routines (http://www.fil.ion.ucl.ac.uk/spm/software/spm8). After motion correction of fMRI images by realignment to the mean image (allowed motion limited to ±4 mm translation and ±3° rotation), the hypothalamus of each participant was manually segmented using the native anatomical image and predefined criteria [57]. Furthermore, a reference region of about the same size as the hypothalamus ROI was drawn in the occipital cortex of each individual. The mean signal of this reference area was subtracted from that in the hypothalamus to correct for global signal drift. To assess differences between conditions, we compared the area under the curve (AUC), which was calculated using the trapezoidal rule by approximating the region under the graph. AUC values were compared using 2-sample paired *t*-tests. The relation between hypothalamus activity and hormonal satiety parameters was assessed using the Pearson product-moment correlation coefficient (2-tailed, *p* < 0.05 corrected for multiple comparisons using Bonferroni correction).

### 4.5. Seed-Based Connectivity Analysis

A whole-brain connectivity analysis, as implemented in the CONN toolbox, version 17 (https://www.nitrc.org/projects/conn [58]), was performed to assess functional coupling between the hypothalamus and the rest of the brain (i.e., voxels contained within the reduced brain mask) in response to glucose and saline infusions. FMRI data were preprocessed using SPM8-based routines (http://www.fil.ion.ucl.ac.uk/spm/software/spm8). To account for magnetic field equilibration, 4 volumes from the start of each functional run were excluded from the analysis. Functional scans were slice-time corrected with reference to the first slice using SPM8′s Fourier phase-shift interpolation. Images were then realigned, with the allowed motion limited to ±4 mm translation and ±3° rotation over the entire experiment. Potential outlier scans were identified using ART [59] as acquisitions with framewise displacement above 0.9 mm or global BOLD signal changes above 5 standard deviations [60], and a reference BOLD image was computed for each subject by averaging all scans excluding outliers. Individual T1 images were co-registered with the mean T2* images and subsequently segmented into grey matter, white matter and cerebrospinal fluid partitions and were spatially normalized to the Montreal Neurological Institute standardized space (http://www.mni.mcgill.ca). The functional volumes were resampled to a 1 × 1 × 1 mm^3^ voxel size and spatially smoothed with a 4 mm full-width half-maximum isotropic Gaussian kernel. In the subsequent seed-to-voxel analysis, the temporal correlation between the BOLD signal from the hypothalamus seed to all other voxels in the brain was computed. In addition, functional data were denoised using a standard denoising pipeline [61], including the regression of potential confounding effects characterized by white matter timeseries (5 CompCor noise components), CSF timeseries (5 CompCor noise components), outlier scans (below 25 factors) [60], session and task effects and their first order derivatives (26 factors) and linear trends (2 factors) within each functional run, followed by bandpass frequency filtering of the BOLD timeseries [62] between 0.008 Hz and 0.09 Hz. CompCor [63,64] noise components within white matter and CSF were estimated by computing the average BOLD signal, as well as the largest principal components orthogonal to the BOLD average and outlier scans within each subject’s eroded segmentation masks.

We then used the individually segmented masks of the hypothalamus as seed masks at the individual level. The time series of both conditions were extracted from each voxel within the ROI in native space. Specifically, we used the realigned fMRI data stemming from the time-series analysis to extract ROI data. This allowed us to extract BOLD data in the original native space of each participant, thereby increasing special fidelity. Block regressors corresponding to the 12 consecutive 2 min time bins were convolved with a canonical hemodynamic response function. For each participant, bivariate Pearson correlation analyses were performed to estimate the connection from the seed hypothalamus ROI to other voxels in the brain using the preprocessed (i.e., normalized and smoothed data) fMRI data. The resulting statistical maps were analyzed in a random-effects group analysis. Specifically, we performed a repeated-measures ANOVA with liquid type and time point as within-subject factors and group as a between-subjects factor. Whole-brain second-level results are reported for the main effect of liquid type. Statistical inference was based on a significance threshold of *p* < 0.05 corrected for multiple comparisons using FWE correction for small volumes and a cluster-defining threshold of *k* > 30.

### 4.6. Biochemical Analysis of Glucose and Insulin

Two blood samples were taken on each day, shortly before and after the fMRI measurement. Assessment of glucose concentrations was performed at the central laboratory of the University Clinic Heidelberg on a Siemens Advia 2400 device using the hexokinase method. Plasma insulin was measured using commercially available kits from Merck Millipore (Merck KGaA, Darmstadt, Germany). Paired Student’s *t*-tests (2-tailed) were employed to compare pre-post assessments of glucose and insulin. To confirm whether participants were successfully blinded to the nature of the liquid, we performed a Pearson’s χ^2^ test for independence (*p* < 0.05).

## Figures and Tables

**Figure 1 ijms-24-07370-f001:**
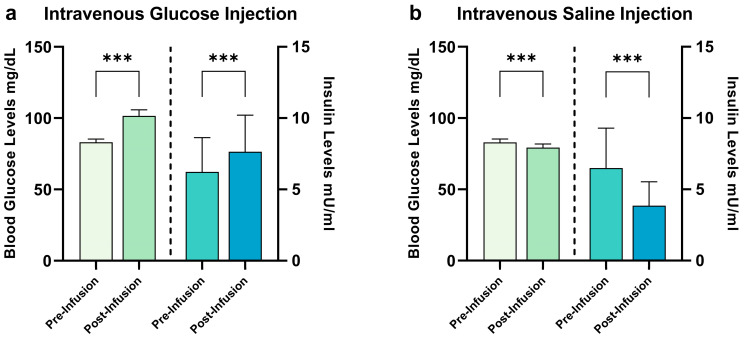
Changes in hormonal satiety parameters following G20 ((**a**): glucose) injection and NaCl ((**b**): saline) injection. During the glucose condition, we observed baseline blood glucose levels of 83.1 ± 6.1 mg/dL and insulin levels of 6.2 ± 2.4 mU/mL. Following glucose infusion, we observed blood glucose levels of 101.5 ± 11.7 mg/dL and insulin levels of 7.7 ± 2.6 mU/mL, a difference of 18.4 ± 12.5 for blood glucose levels (significant at *p* < 0.001) and 1.2 ± 2.6 for insulin levels (*p* < 0.001). During the saline condition, we observed baseline blood glucose levels of 82.9 ± 6.6 mg/dL and insulin levels of 6.5 ± 2.8 mU/mL. Following saline infusion, we observed blood glucose levels of 79.3 ± 15.7 mg/dL and insulin levels of 3.8 ± 1.7 mU/mL, a difference of −6.2 ± 15.5 for blood glucose levels (*p* < 0.001) and −2.6 ± 1.8 for insulin levels (*p* < 0.001). *** *p* ≤ 0.001, paired Student’s *t*-test, bars depict mean with SD.

**Figure 2 ijms-24-07370-f002:**
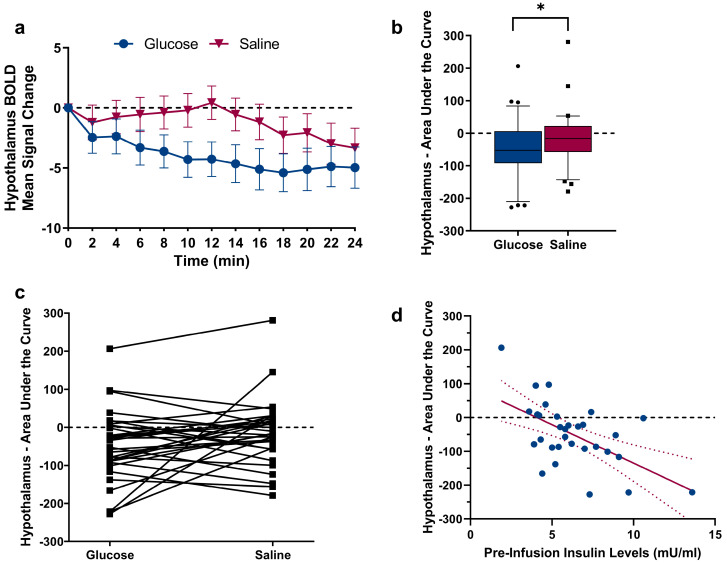
(**a**) BOLD signal change in the hypothalamus Region-Of-Interest (manually segmented on individual native space brain scans) following glucose and saline infusion over time. (**b**) A comparison of the AUC revealed significant differences between glucose and saline (t_30_ = 1.88, *p* = 0.035). * *p* ≤ 0.05, paired Student’s *t*-test. Individual data points above and below the 10th and 90th percentiles are shown as squares and dots. (**c**) Data points for each participant plotted with lines showing the difference in AUC between glucose and saline infusion. (**d**) Negative correlation between blood plasma insulin levels assessed before MRI scanning and hypothalamic AUC following glucose injection (r = −0.579, *p* < 0.001), Pearson product-moment correlation coefficient (2-tailed). Regression line is depicted with 95% confidence bands.

**Figure 3 ijms-24-07370-f003:**
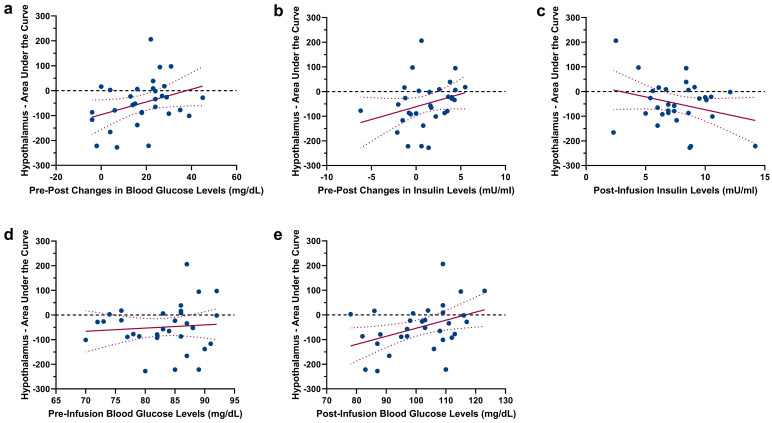
Scatterplots showing the non-significant association between the AUC of hypothalamic reactivity following glucose injection and (**a**) changes in blood glucose levels (pre-infusion levels vs. post-infusion levels), (**b**) changes in insulin levels (pre-infusion levels vs. post-infusion levels), (**c**) post-infusion insulin levels, (**d**) pre-infusion blood glucose levels and (**e**) post-infusion blood glucose levels. Regression lines are depicted with 95% confidence bands.

**Figure 4 ijms-24-07370-f004:**
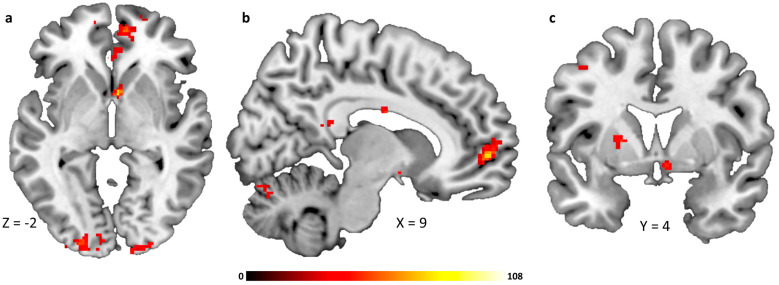
Brain regions showing glycemia-dependent functional connectivity with the hypothalamus, (**a**) medial prefrontal/orbitofrontal cortex and caudate head, (**b**) medial prefrontal/orbitofrontal cortex, (**c**) putamen. All whole-brain results are reported at a threshold of *p* < 0.05 corrected for multiple comparisons using FWE correction for small volumes and a cluster-defining threshold of *k* > 30 voxel minimal cluster size. Color bar indicates *F*-values.

**Figure 5 ijms-24-07370-f005:**
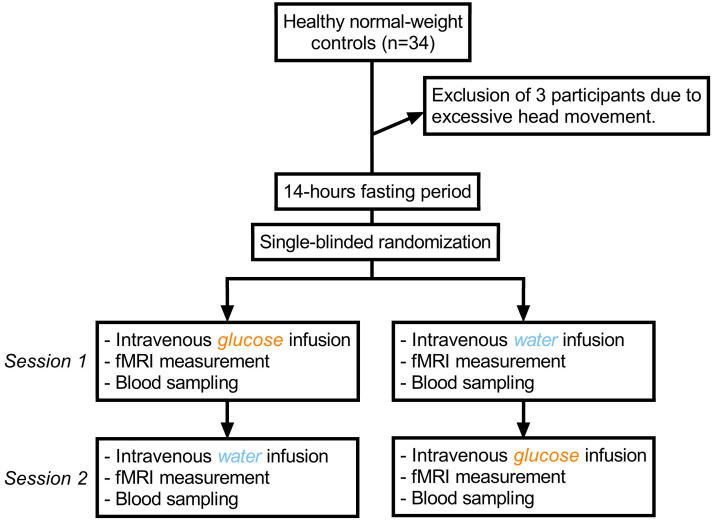
Graphical depiction of the study protocol.

**Table 1 ijms-24-07370-t001:** Participant characteristics.

	Mean ± SD (MIN–MAX)
Female/male	17/14 (N = 31)
Age	24.1 ± 4.7
BMI (kg/m^2^)	21.9 ± 1.8
EDE-Q	0.3 ± 0.4 (0–1.9)
BDI-II	2.5 ± 2.6 (0–11)
MWT-B	30.1 ± 3.2 (24–36)

EDE-Q: Eating Disorder Examination Questionnaire, BDI-II: Beck Depression Inventory, MWT-B: Vocabulary-Based Test for the Assessment of Premorbid Intelligence.

**Table 2 ijms-24-07370-t002:** Glycemia-dependent functional connectivity of the hypothalamus.

Region	Hemisphere	*x*	*y*	*z*	*k*	*F*-Value	*p*-Value
Anterior Cerebellum	R	22	−32	−24	39	108	<0.001
Parahippocampal Gyrus	R	14	0	−16	79	83	Id.
Caudate Head	R	4	8	−2	Id.	75	Id.
Medial Frontal/Orbitofrontal Cortex	R	8	54	0	165	78	Id.
Middle Occipital Gyrus	R	34	−94	−6	220	76	Id.
Posterior Cerebellum	R	6	−76	−16	36	76	Id.
Anterior Cingulate Cortex	R	2	40	−6	54	74	Id.
Middle Occipital Gyrus	L	−26	−100	2	218	72	Id.
Hippocampus	R	40	−14	−20	30	71	Id.
Fusiform Gyrus	L	−36	−74	−18	58	67	Id.
Posterior Cingulate Cortex	R	4	−50	14	77	66	Id.
Precuneus	L	−8	−64	42	23	56	Id.
Superior Temporal Gyrus	L	−42	−34	12	27	55	Id.
Ansiform Lobule	R	18	−86	−20	26	54	Id.
Anterior Cerebellum	R	24	−46	−28	28	52	Id.
Vermis	R	32	−70	−40	49	51	Id.
Anterior Medial Prefrontal Cortex	L	−8	52	2	45	50	Id.
Inferior Occipital Gyrus	R	10	−100	−10	33	50	Id.
Inferior Parietal Lobule	R	40	−42	52	30	50	Id.
Putamen	L	−20	4	6	35	50	Id.
Precentral Gyrus	L	−44	−12	48	57	44	Id.

All whole-brain results are reported at a threshold of *p* < 0.05 corrected for multiple comparisons using FWE correction for small volumes and a cluster-defining threshold of *k* > 20 voxel minimal cluster size. L = left; R = right; *k* = number of voxels.

## Data Availability

The data presented in this study are available on request from the corresponding author. The data are not publicly available due to ethical reasons.

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
