# Peer review of "Hypothalamic Reactivity and Connectivity following Intravenous Glucose Administration"

_ijms, 2023, doi:10.3390/ijms24087370_

Round 1

Reviewer 1 Report (Previous Reviewer 1)

No further comments

Author Response

We thank the reviewer for endorsing our manuscript. 

Reviewer 2 Report (New Reviewer)

Accurate glucose sensing is critical for animals to maintain energy homeostasis, and the dysfunction in this process can lead to detrimental diseases like obesity and type 2 diabetes. In this manuscript, the authors used fMRI to investigate how key brain regions involved in glucose sensing respond to intravenous infusion of glucose. With rigorous experimental design, the authors found that hypothalamic activity is decreased after infusion of glucose within physiologically relevant concentration range. Interestingly, they also discovered that the reactivity is negatively correlated with pre-infusion insulin levels, but not changes in blood glucose level. They further showed evidence suggesting that reward-related brain regions showed similar activity patterns with hypothalamus, supporting a functional connectivity during glucose sensing.

Overall, this manuscript provides convincing evidence to support the main conclusions. Experimental data is clearly and accurately presented. Additionally, the use of intravenous infusion to manipulate blood glucose levels while bypassing oral and gastrointestinal effects provides critical insights into our understanding of glucose sensing in the brain. However, before I can recommend publication, the following comments should be addressed by the authors first.

1. In the text in Line 98, the authors reported “a mean dose of 9175mg”, while in Table 2, the mean infusion dose is 9109.7 mg. Why is there a discrepancy?

2. The authors should plot the data in Table 2 for better visualization. Additionally, the authors should describe what statistical analysis was done to calculate p-value in the figure legend.

3. Line 107. The author should briefly explain what “BOLD” represents and how to interpret its value somewhere in the manuscript for a broader audience. Right now it is not described. Also, this subtitle is confusing since the authors showed in Figure 1 that glucose actually decreased hypothalamus activity while saline does not have an effect.

4. In Figure 1b, it would be great if the authors could indicate which dots are from the same individual so that it is easier to see the difference between glucose and saline with each individual. Currently this critical information is not presented in the plot.

5. Line 118: “pre-glucose” is somewhat confusing in this setting. The authors can use “pre-infusion insulin” just as what is labeled on the plot in Figure 1c.

6. Line 119-121: I don’t think the BMI data is presented anywhere. The authors should show this data as well.

7. For Figure 3, in addition to the activity maps shown, the authors should also present BOLD/AUC values for these regions with glucose versus saline infusion.

8. In the discussion, the authors compared the effects of intragastric versus intravenous infusion of glucose in great detail, and provides intriguing hypothesis explaining the discrepancy observed with these two methods. I wonder if the authors could perform intragastric infusion with the exact same dose as intravenous infusion, and then make direct comparisons without any ambiguity. This will provide fundamental insights into the mechanisms of glucose sensing, and shed light on whether pre-absorptive versus post-absorptive glucose is sensed differently.

Round 2

Reviewer 2 Report (New Reviewer)

The authors have addressed all my concerns and I strongly recommend the manuscript for publication.

This manuscript is a resubmission of an earlier submission. The following is a list of the peer review reports and author responses from that submission.

Round 1

Reviewer 1 Report

General comment:

This is an interesting study that have assessed the hypothalamus responses to glucose samples administered intravenously in healthy normal weight participants. The study used physiological fMRI paradigm to assess hypothalamus BOLD responses and functional connectivity analyses. The study has some methodological issues, particularly in data analysis and subsequently could have affected the results. These issues are addressed below

Abstract:

·       Line 16: the authors refer in this manuscript that data were acquired at high resolution, however images were acquired at 2mm isotropic, which is not particularly high resolution in recent days. This resolution is nowadays typically used in most recent fMRI studies as a conventional voxel size. I suggest the authors only indicate the voxel size in the method section and take out the high-resolution from the rest of the manuscript.

·       line 18: as authors have not directly measured satiety responses either through questionnaire or gut peptides, it would make more sense that authors refer to the hypothalamus responses as glucose/ glycaemia-dependent functional connectivity rather than satiety-dependent. In addition, this was based on the correlation between insulin and hypothalamus responses pre-injection. Could the authors expand more why they think it is a satiety sensing? Please, change satiety throughout the manuscript to glucose/ glycaemia-dependent.

·       line 20/21: the authors indicate that “The observed effect size was smaller than during oral or intragastric administration of glucose, demonstrating the important role of the digestive process in homeostatic signalling”. This sentence needs rephrasing as currently indicates that the authors have performed oral and intragastric administration in the current study in addition to the intravenous sample injection, which they have performed. Perhaps need to say compared to previous studies to make it more clear

·       Line 23: the authors highlighted that the administered glucose sample is of a small amount/concentration, was this the aim of the study, i.e. what make the study different from previous work?

Introduction:

 Line 83: what HT stands for, first time mentioned in the manuscript

Results

2.2 Glucose and saline induced BOLD activation in the hypothalamus.

·      Line 114: I suggest rephrasing “we failed to observe xxx” to No significant correlation between AUC values and changes in blood glucose levels was found. Also it is unclear to me whether this insignificant correlation was for saline or glucose ?

·      Line 119: it is unclear how the correlation between insulin and hypothalamus AUC was performed? Was this done for pre-infusion data of both visits combined? if not then which visit, glucose or saline?. Also the authors indicate that “the correlation is pointing towards a relation between increased hypothalamic-reactivity and increased preprandial blood plasma insulin levels”. If this proves to be correct, also see comment regarding this point in the abstract,  I suggest this to be phrased as relation between decreased hypothalamic activity and increased preprandial blood plasma insulin levels. On the same note, I cannot see the point of correlating pre-infusion insulin levels with hypothalamus responses, could the authors explain their rational?

·      Line 122: it is unclear to me why authors correlated BMI of normal weight participants with blood measurements. I do not think this would give any meaningful data or corelation.

·      Figure 1: the study comparing glucose vs saline, why legends indicating glucose vs water. Y-axis in the middle and right graph should indicate that AUC is for hypothalamus.  

Discussion: Line155: please see previous comment of the corelation.

Method:

4.1 Study design and test group

·      Line 253. The authors indicate that one subject with BMI>25 Kg/m2 was included in the study based on analysis/measurement made resulted that weight caused by increased muscle mass and not fat. How was this measured? Also was fat/muscle ratio measured for all participants? I suggest removing this participant from the analysis and have a more homogenous cohort.  

·      Line 254, It is unclear to me why participants were assessed for their vision?

4.2 Procedure

·      Adding a diagram of the protocol will be beneficial  

·      Could the author explain why they have not administer a standardised dose of glucose and control, given that all participants are normal weight (apart from the included volunteer with BMI>25)?. Also according to the authors, calculation of the intervention does was based on a rough estimate of glucose levels, which they have not used in their further analysis, can they explain in this further?

·      See previous comments on the high-resolution acquisition  

4.4 fMRI analysis

·      Line 316, correct of grammar, please use was compared with the baseline xxx instead of will compared with xxx

·      What software was used for the fMRI analysis, also did the authors perform slice timing correction?

·      The authors chose 4mm as their cut-off to motion limit, this is 2 times the acquired voxel size. A role of thumb motion correction should be more a voxel size, also taken into account the small size of the hypothalamus this is considered to be a big allowance for motion. Can authors comment on this?

4.5 Seed-based connectivity analysis.

·      Line 345: I am surprised that the authors chose a spatial smoothing of 8mm. This very heavy smoothing does not align with the acquired voxel size and the previous pre-processing steps. The acquired voxel size is 2mm, and data was resampled during normalisation process to 1mm isotropic. I suggest the authors use 4mm smoothing (double the acquired voxel size) or explain why they have chosen 8mm smoothing.

·      Line 351: how was physiological noise corrected for? Was this recorded during scanning and then corrected, or did authors applied high pass-filter during analysis, if so what was the cut-off used?

·      Line 367: authors indicated cluster threshold of k>30 while it is reported as k>20 in figure 2

Reviewer 2 Report

The authors use fMRI to investigate how the hypothalamus reacts to intravenous glucose administration. They also investigate the functional connectivity between the hypothalamus and image voxels representing the whole brain. They found a decrease in hypothalamic activity following a small amount of glucose administration and a significant glucose-dependent connectivity pattern between the hypothalamus and reward-related brain regions. The manuscript is very clearly written and the results of a significant hypothalamic reactivity to low dose glucose infusion are of scientific interest to a wider field of research.

Point-to-point comments

ABSTRACT

Lines 19–20: The statement “a hypothalamic response to glucose infusion which was inversely related to fasting insulin levels” makes no sense if the reader is unaware of the fact that hypothalamic reactivity is decreased following glucose administration. Consider rephrasing.

INTRODUCTION

Lines 57–70: The paragraph is somewhat vague. You write that (pato)physiology homeostatic regulations remain “insufficiently understood”. What are the next steps to understand the physiology better? Can you be more to the point? What do you mean by “meaningful inferences”?

RESULTS

Lines 114–121: Regarding the results from the correlations between AUC and insulin/glucose levels, I would like to see all results presented in a graph like Figure 1, right. With such graphs the reader can judge trends and spread of data. You did quite a few correlations between AUC and differences/pre-scanning/post-scanning levels of glucose and insulin. Did you correct for multiple comparisons? Are the results significant after correction?

Lines 126–128: In Figure 1 left and middle the label in the figure is “water” but in the caption and in the text, it is “saline”. Please use consequent terminology.

Line 139: Table 3. Here you state that the cluster threshold is k>20, but in the Methods section, you write k>30. Which threshold is correct?

DISCUSSION

Line 155: Regarding “significant correlation”, see comment to Lines 114–121.

MATERIALS AND METHODS

Line 254: How did you measure/assess “increased muscle mass”?

Lines 266–268: Please give range data for the questionnaires and cut-off values for clinically significant impairment. This information is necessary for the reader to interpret Table 1.

Line 316: Possible typing error: “will”.

Line 367: See comment to Line 139.

Reviewer 3 Report

The work of Simon and colleagues focuses on understanding glucose signaling to the brain by assessing its reactivity in the hypothalamus and its interaction with mesocorticolimbic regions of the brain, through intravenous administration in 31 healthy, normopeso participants of glucose and saline and subsequent high-resolution fMRI analysis.

General comment. The work although interesting, because it allows the study of glucose signaling independently of digestive processes, in this form would be better presented for brief communication. In fact, the results are meager. Moreover, as the authors themselves afferent, the limitations of the work are numerous. It seems almost like an appendix to the authors' own plucited work 'Neuroimaging of hypothalamic mechanisms related to glucose metabolism in anorexia nervosa and obesity' (Simon et al., 2020). Instead, it might be interesting for the authors to open up to gender dimorphism, to investigate more about different hypothalamic nuclei.

Minor comments.
Introduction. The last sentence should be moved to the discussion.
Results.
-interesting would be to break down the data by gender.
- Detail the acronym AUC
- In the text, insert references to individual portions of the figures
-Fig 1. Attention typo.
- Fig 1-2. Insert letters as references to individual portions of the figures.
Methods.
- Could the authors explain the choice of the dose of glucose administered (G20)?
- Was the experiment always conducted in the same time slot?
- Could a more robust statistical evaluation be used (1 way ANOVA for example).
